# ECHORAG: A COGNITIVE MEMORY-INSPIRED FRAMEWORK FOR RAG WITH SEMANTIC GIST

## ABSTRACT

Retrieval-Augmented Generation (RAG), a pivotal technology connecting external knowledge with large language models, has been widely applied in various knowledge-intensive tasks. However, due to the inherent discrete representation of textual information and retrieval paradigms in current mainstream RAG systems, there is a prevalent issue of lack of semantic integrity, which leads to deviations in semantic retrieval. Therefore, we propose the concept of semantic gist and design EchoRAG, a novel RAG framework that simulates human cognitive memory. Specifically, inspired by the human episodic memory mechanism, this framework first achieves an understanding of semantic gist through reasoning and uses this to construct a multi-dimensional knowledge graph. During retrieval, we introduce the Cognitive Diffusion Module. This unified mechanism performs global thinking across the knowledge graph, guided by structural relevance and entity frequency. It simultaneously constructs a holistic semantic landscape and evaluates node significance via our proposed CogniRank algorithm, which integrates semantic-space relevance with graph-topological importance to determine the final passage ranking. To verify the effectiveness of EchoRAG, experiments were conducted on 5 public datasets for Question Answering (QA) tasks and multi-hop reasoning tasks. The results show that compared with current mainstream RAG methods, EchoRAG significantly improves answer accuracy and recall metrics while enhancing speed.

## 1 INTRODUCTION

Retrieval-Augmented Generation (RAG), as a key technique for connecting external knowledge with Large Language Models (LLMs), has become the dominant paradigm by dynamically retrieving relevant information to alleviate model hallucinations (Lewis et al., 2020; Guu et al., 2020a). A typical RAG system generally follows a three-step pipeline: knowledge construction, retrieval, and reranking (Gao et al., 2023).

To structure the knowledge base in the construction phase, recent efforts have shifted from simple text indexing (Lewis et al., 2020; Guu et al., 2020b; Borgeaud et al., 2022) to building structured knowledge graphs (Saleh et al., 2024; Huang et al., 2025; Jiang et al., 2025; Guo et al., 2024). However, this initial stage is, in essence, a lossy compression process (Li et al., 2024; Zhu et al., 2025), which discards substantial context and narrative background, leading to a deficit in comprehension and memory. During the retrieval stage, these deficiencies are further magnified. Even with the introduction of advanced iterative (Wang et al., 2025) or agentic (Maragheh et al., 2025; Ravuru et al., 2024) reasoning frameworks, these systems can comprehend the associations between entities, they fail to understand how these associations collectively constitute a meaningful semantic scene, thereby becoming trapped in localized reasoning (Fountas et al., 2024; Gutiérrez et al., 2025). While advanced reranking techniques exist (Sun et al., 2025; Yang et al., 2025; Yu et al., 2024; Jin et al., 2025), they remain largely focused on evaluating the local relationship between a query and its candidate documents, failing to fundamentally resolve the limitations originating from flawed knowledge base construction and localized retrieval, as illustrated in Figure 1.

In contrast to the localized reasoning of current systems, the human brain seamlessly integrates vast amounts of knowledge. This remarkable ability is explained by well-established cognitive theories of memory. Foundational research distinguishes memory for verbatim details from "gist memory", a more durable understanding of semantic essence (Reyna & Brainerd, 1995). Furthermore, "episodic

memory" allows for the holistic comprehension of information by situating it within a coherent spatiotemporal scene (Tulving et al., 1972). During recall, the brain then employs "importance judgment" to rapidly identify core concepts from this landscape (Kintsch & Van Dijk, 1978).

Inspired by this, in this work, we propose EchoRAG, a novel RAG framework that mimics and operationalizes these core cognitive processes across a three-stage pipeline. To emulate gist memory, our framework begins by distilling the semantic gist of knowledge through reasoning, constructing a multi-dimensional knowledge graph around this essence to preserve rich context. Building on this foundation, we introduce the Cognitive Diffusion Module, a unified mechanism operationalizing both episodic memory and importance judgment. This module propagates relevance throughout the graph, guided by each node's structural relevance and entity frequency. The process culminates in our proposed CogniRank algorithm, which reranks knowledge by integrating both semantic-space relevance and graph-topological importance through a weighted fusion strategy. This allows it to simultaneously construct a context-aware scene while calculating global concept importance, with the final diffusion scores directly determining the knowledge ranking.

Overall, the main contributions of this paper are as follows:

(1) We propose the concept of semantic gist and EchoRAG, a novel RAG framework that simulates human cognitive memory to address semantic integrity loss by constructing a multi-dimensional knowledge graph.

(2) We design a Cognitive Diffusion Module and the CogniRank algorithm, which emulate episodic memory and importance judgment to achieve global, context-aware reranking of knowledge passages.

(3) Extensive experiments on five benchmarks demonstrate that EchoRAG significantly outperforms state-of-the-art methods in accuracy, recall, and efficiency on QA and multi-hop reasoning tasks.

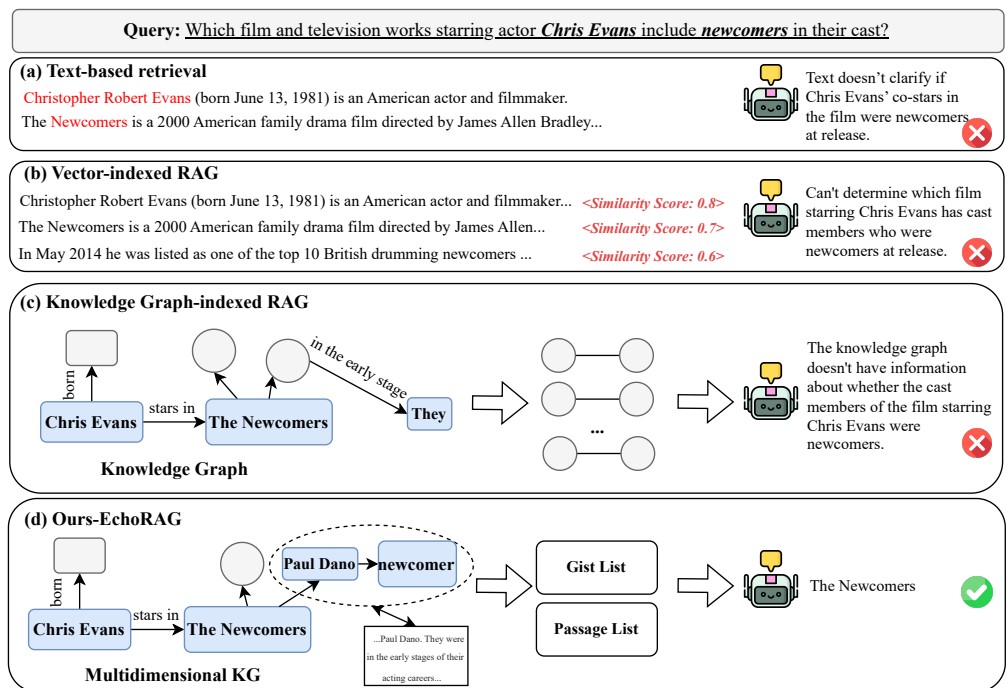

Figure 1: The illustration shows the differences among four retrieval methods: text based retrieval, vector indexed RAG, knowledge graph indexed RAG, and EchoRAG, in an example query.

## 2 RELATED WORK

**Retrieval-Augmented Generation** Retrieval-Augmented Generation (RAG) improves Large Language Models (LLMs) by grounding them in external knowledge, which helps reduce hallucinations and improve factual accuracy (Lewis et al., 2020; Jiang et al., 2023). However, the standard RAG paradigm, which decomposes documents into independent chunks, causing the retrieval process to fall into a local optima trap where it retrieves semantically related but contextually incomplete fragments (Barnett et al., 2024). While advanced frameworks have sought to address this, they often do so through sequential, multi-step processes. For instance, iterative architectures like ITER-RETGEN (Shao et al., 2023) were developed for multi-hop reasoning by progressively refining queries. Similarly, methods like Self-RAG (Asai et al., 2024) introduce self-reflection and critique loops to validate retrieved context. While effective, these iterative and reflective steps approach reasoning as a linear chain, which fail to capture the holistic semantic gist required for complex problems. In contrast, EchoRAG is designed to overcome this fundamental limitation by first establishing a global semantic landscape, ensuring that the subsequent retrieval is guided by a comprehensive understanding rather than localized signal matching.

**Graph Retrieval-Augmented Generation** To address the limitations of processing linear text for complex tasks, recent research has gravitated towards incorporating graph-structured knowledge. Some methods, like KAPING (Baek et al., 2023), retrieve knowledge graph (KG) subgraphs and flatten them into text, but this loses crucial relational context. Other approaches, such as ToG (Sun et al., 2023), empower an LLM agent to perform symbolic reasoning by traversing the graph. This agent-based traversal, however, can still be considered a form of localized, step-by-step reasoning within the graph. This evolution from linear chains to graph-based structures in external knowledge retrieval mirrors a parallel shift in the LLM's internal reasoning patterns, from the linear Chain-of-Thought (CoT) to the non-linear Graph-of-Thoughts (GoT), which allows for the synthesis of information from multiple reasoning paths (Besta et al., 2025).

This trend highlights a move toward architectures with more integrated and holistic reasoning capabilities. Brain-inspired models like HippoRAG (Wang et al., 2024a) simulates the human memory system by converting corpora into a KG and using an associative retrieval algorithm. Its successor, HippoRAG-2 Gutiérrez et al. (2025), refines this model by integrating conceptual and contextual information. In contrast, EchoRAG advances this frontier with a novel framework inspired. During the retrieval process, It employs a Cognitive Diffusion Module to perform global thinking across the entire knowledge graph, constructing a comprehensive semantic landscape and employs the CogniRank algorithm to efficiently rank candidate knowledge. This allows EchoRAG to identify the most salient knowledge from a global perspective in a single, unified process, moving beyond the local, sequential reasoning steps that characterize previous methods and achieving a more profound level of semantic integrity.

## 3 PRELIMINARIES

**Retrieval-Augmented Generation (RAG).** Retrieval-Augmented Generation (Lewis et al., 2020; Guu et al., 2020a) is a widely adopted framework that integrates retrieval with generation to enhance the factuality and grounding of large language models (LLMs). Given a user query $q$, a retriever module $R$ identifies $k$ relevant documents $D_q = \{d_i\}_{i=1}^k$ from a large corpus $\mathcal{D} = \{d_i\}_{i=1}^N$, based on vector similarity. Formally:

$$D_q = \arg\text{top-k}\left\{ \frac{E_d(d_i)^\top \cdot E_q(q)}{\|E_d(d_i)\|\|E_q(q)\|} \mid d_i \in \mathcal{D} \right\} \tag{1}$$

where $E_d(\cdot)$ and $E_q(\cdot)$ denote the embedding functions for documents and the query, respectively.

The retrieved documents are concatenated with the query and passed to the LLM reader $\pi_\theta$ to generate a response:

$$y \sim \pi_\theta(y \mid q, D_q) \tag{2}$$

**Instruction-Following in RAG.** Modern RAG systems often operate under instruction-following settings (Sun et al., 2023; 2024; Wang et al., 2024a; Gutiérrez et al., 2025), where users provide explicit task directives $I = \{I_j\}_{j=1}^M$ along with the query $q$. The LLM $\pi_\theta$ is expected to generate

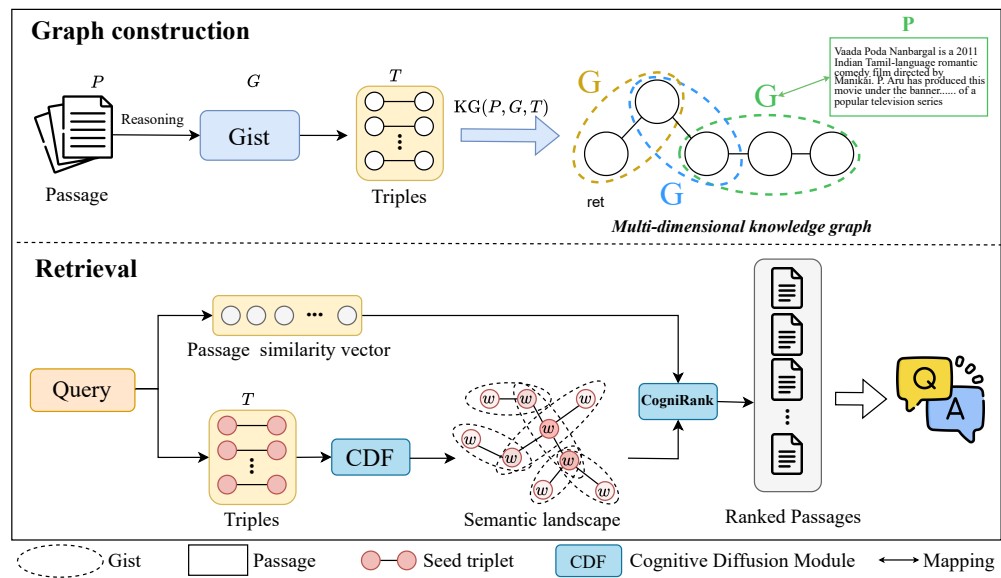

Figure 2: Overview of EchoRAG's two-stage workflow: constructing a multi-dimensional knowledge graph from passages via semantic gist, then retrieving and ranking passages using the Cognitive Diffusion Module and CogniRank for query answering.

output $y$ that not only answers the query using evidence from $D_q$, but also satisfies the constraints described in $I$:

$$y \sim \pi_\theta(y \mid q, D_q, I) \tag{3}$$

**Limitations of Standard RAG.** Despite its effectiveness, standard RAG pipelines suffer from limitations in long-term factual retention, semantic coherence, and disambiguation due to noisy or redundant retrieval. These limitations motivate the need for structured and cognitively inspired memory augmentation techniques, such as **EchoRAG**, introduced in the next section.

## 4 METHOD

### 4.1 OVERVIEW

As shown in Figure 2, EchoRAG implements a retrieval-augmented generation framework that simulates human cognitive memory. It addresses the lack of semantic integrity in conventional RAG systems by reorganizing textual knowledge into a multi-dimensional knowledge graph and enabling graph-based reasoning. This design is inspired by the way the human brain encodes, associates, and recalls information: first capturing the semantic *gist* of knowledge (Reyna & Brainerd, 1995), then situating it within a coherent episodic scene (Tulving et al., 1972), and finally applying an importance judgment mechanism (Kintsch & Van Dijk, 1978).

Formally, given a document collection $D = \{d_i\}$ and a user query $q$, EchoRAG proceeds in two stages: (1) *offline indexing*, which parallels memory consolidation by transforming unstructured corpora into a graph integrating entities, relational facts, and passages; and (2) *online retrieval*, which mirrors human recall by diffusing activation across the graph and refining importance judgments via the proposed COGNIRANK algorithm. The final retrieved knowledge is denoted as $\mathcal{T}(q)$, which serves as input to the generator $\pi_\theta$ for answer synthesis:

$$y = \pi_\theta(q, \mathcal{T}(q)) \tag{4}$$

## 4.2 OFFLINE INDEXING

The offline stage parallels the *memory consolidation process* in humans, where raw sensory experiences are reorganized into structured multi-level representations for long-term storage. This results in a *multi-dimensional knowledge graph* $\mathcal{G}$ that integrates entity nodes, relation types, grounded facts (triples), and passage nodes, thereby supporting compositional and global reasoning. By distilling the semantic *gist* (Reyna & Brainerd, 1995) rather than storing verbatim text, the system prevents semantic fragmentation and preserves the narrative background.

Formally, we build $\mathcal{G} = (\mathcal{V}, \mathcal{E}, \mathcal{F}, \mathcal{P})$, where $\mathcal{V}$ are entity nodes, $\mathcal{E}$ are relation types, $\mathcal{F}$ are factual triples (facts), and $\mathcal{P} = \{p_j\}$ are passages.

**Document Segmentation.** Each document $d_i \in D$ is segmented into semantically coherent passages $p_j$, forming a passage set $\mathcal{P} = \{p_j\}$. Analogous to how the hippocampus stores episodic fragments (Tulving et al., 1972), passages provide local textual grounding that anchors entities and relations within broader semantic episodes.

**Semantic Gist Extraction.** For each passage $p$, we extract a set of semantic gists $\mathcal{GI} = \{g(p_i)\}$. A gist is defined as a concise abstraction that captures the central proposition of the passage while discarding surface-level lexical details. Operationally, we implement gist extraction using LLM same as the answer model with the following prompt: "Summarize the passage in one sentence that captures its key meaning without copying phrases." Multiple gists can be generated per passage, and they are linked as auxiliary nodes in $\mathcal{G}$. This procedure ensures that "gist" refers to a reproducible text unit rather than an informal intuition.

**Entity and Fact Extraction.** From each gist $g \in \mathcal{GI}$, an information extraction model $f_{\text{rel}}$ identifies entities and relational triples $t = (v_h, r, v_t)$, $v_h, v_t \in \mathcal{V}$, $r \in \mathcal{E}$. Each triple is grounded to its supporting gist $g$, preserving episodic context. To facilitate later retrieval, we also represent each triple as a textualized fact $f = \text{concat}(v_h, r, v_t)$, $f \in \mathcal{F}$. This dual representation (structured graph + textual string) parallels the coexistence of abstract semantic schemas and detailed episodic traces in human memory (Reyna & Brainerd, 1995; Tulving et al., 1972).

**Multi-dimensional Graph Construction.** The constructed graph links entities through relation types and connects facts and passages as grounded evidence. This mirrors how human memory situates discrete events within a spatiotemporal scene (Tulving et al., 1972). Edges in the entity graph are induced from $\mathcal{F}$, and passages are attached to the entities and facts they contain: $\mathcal{G} = (\mathcal{V}, \mathcal{E}, \mathcal{F}, \mathcal{P})$, $(v_h \xrightarrow{r} v_t)$ for each $(v_h, r, v_t) \in \mathcal{F}$.

**Vector Encoding.** To enable similarity-based reasoning, all objects are embedded into a shared vector space:

$$e_v = E_v(v), \quad v \in \mathcal{V}, \tag{5}$$
$$e_r = E_r(r), \quad r \in \mathcal{E}, \tag{6}$$
$$e_f = E_f(\text{concat}(v_h, r, v_t)), \quad f \in \mathcal{F}, \tag{7}$$
$$e_p = E_p(p), \quad p \in \mathcal{P}, \tag{8}$$
$$e_q = E_q(q). \tag{9}$$

These embeddings are analogous to distributed neural codes in human cognition, where both semantic gist and episodic traces are jointly represented (Reyna & Brainerd, 1995).

## 4.3 ONLINE RETRIEVAL

The online retrieval stage simulates the *recall process*, where a query triggers associative activation across related memory traces (Kintsch & Van Dijk, 1978). EchoRAG combines local similarity signals (analogous to direct cue matching) with global diffusion (analogous to spreading activation across a semantic network), and incorporates frequency signals as a proxy for importance judgment.

### 4.3.1 SIMILARITY COMPUTATION

The query is encoded: $e_q = E_q(q)$. Cosine similarity is computed against all facts and passages: $\sigma(q,x) = \frac{\langle e_q, e_x \rangle}{\|e_q\|\|e_x\|}$, $x \in \mathcal{F} \cup \mathcal{P}$. Top-$K$ facts are selected by $\sigma(q,f)$: $\mathcal{F}_K(q) = \text{TopK}_{f \in \mathcal{F}}(\sigma(q,f))$. These correspond to the most *directly activated* memory traces in response to the query.

### 4.3.2 COGNIRANK DIFFUSION

Each entity node $v$ is initialized with a fused score that combines two components:

(1) **Fact-based similarity.** Instead of computing similarity directly between the query and an isolated entity, we assess entity relevance through the facts in which it is embedded. This design mimics human cognition: people typically recall entities via their participation in relational contexts (e.g., facts, episodes) rather than as disconnected symbols. By averaging the query–fact similarities for all top-$K$ facts that mention $v$, we capture not only the direct association between the query and $v$, but also the relational structure among entities, analogous to how neurons encode associations through interconnected pathways:

$$\text{sim}(v) = \frac{1}{|\{f \in \mathcal{F}_K(q) : v \in f\}|} \sum_{f \in \mathcal{F}_K(q)} \mathbf{1}[v \in f] \cdot \sigma(q,f), \tag{10}$$

where $\sigma(q,f)$ denotes the cosine similarity between the query $q$ and fact $f$.

(2) **Entity-frequency reward.** Similar to human importance judgment (Kintsch & Van Dijk, 1978), frequent co-occurrence signals that an entity plays a central role within the retrieved context. So entities that occur more frequently in $\mathcal{F}_K(q)$ are rewarded. Let $c_v$ denote the hit count of $v$ in the top-$K$ facts. The reward is defined as:

$$\text{reward}(v) = 1 + \alpha\big(1 - e^{-\beta c_v}\big), \tag{11}$$

where $\alpha$ controls the strength of the reward and $\beta$ controls the saturation rate.

The initial activation for node $v$ is then given by a weighted sum of fact-based similarity and frequency reward:

$$\pi_0(v) = \lambda_1 \cdot \text{sim}(v) + \lambda_2 \cdot \text{reward}(v), \tag{12}$$

where $\lambda_1$ and $\lambda_2$ are hyperparameters balancing semantic similarity and frequency-based importance.

CogniRank then simulates spreading activation in semantic memory (Tulving et al., 1972). Starting from $\pi_0$, activation diffuses through the graph according to an iterative update rule:

$$\pi_{t+1} = (1-\gamma)W^\top \pi_t + \gamma \pi_0, \tag{13}$$

where $\pi_t \in \mathbb{R}^{|\mathcal{V}|}$ is the activation vector at step $t$, and $\gamma \in (0,1)$ is a restart probability that controls the balance between global diffusion and returning to the initial distribution $\pi_0$. A larger $\gamma$ biases the process toward local evidence from $\pi_0$, while a smaller $\gamma$ allows activation to diffuse more broadly across the graph.

The matrix $W \in \mathbb{R}^{|\mathcal{V}| \times |\mathcal{V}|}$ is the column-normalized adjacency matrix of the entity-fact graph. More precisely, if there is an edge $(u,v) \in \mathcal{E}$, then

$$W_{uv} = \frac{1}{\deg(v)}, \tag{14}$$

where $\deg(v)$ is the degree of node $v$. This ensures that each node evenly distributes its activation to its neighbors during the diffusion process. Intuitively, $W^\top \pi_t$ corresponds to a single step of probability mass propagation across the graph, while the interpolation with $\pi_0$ anchors the diffusion to entities that were directly supported by retrieved facts.

The iterative process converges to a stationary distribution $\pi^*$, which captures both the direct importance of entities (through fact similarity and frequency reward) and their contextual relevance via global associations. This distribution serves as the basis for downstream passage scoring.

### 4.3.3 PASSAGE RERANKING

For each passage $p$, CogniRank produces an entity-based score by taking the maximum activation among its entities:

$$S_{\text{cog}}(p \mid q) = \max_{v \in \mathcal{V}(p)} \pi(v). \tag{15}$$

To incorporate direct cue matching, we also compute passage-query similarity $\sigma(q, p)$. The final passage score is a weighted combination:

$$S(p \mid q) = \lambda_3 \cdot S_{\text{cog}}(p \mid q) + \lambda_4 \cdot \sigma(q, p). \tag{16}$$

Passages are reranked by $S(p \mid q)$, so that those containing globally important entities and strong direct similarity emerge as top-ranked. Analogous to human recall (Reyna & Brainerd, 1995; Kintsch & Van Dijk, 1978), central details are prioritized over peripheral ones. These passages, along with their supporting entities, form the retrieved knowledge $\mathcal{T}(q)$.

## 5 EXPERIMENT

### 5.1 EXPERIMENTAL SETUP

#### 5.1.1 BASELINE

We compare EchoRAG with several well-established retrieval methods and structure-enhanced RAG systems, including: (1) None, a QA model without external retrieval, relying solely on parametric knowledge; (2) NV-Embed-v2 (7B) (Lee et al., 2024), a strong dense retrieval baseline; (3) GraphRAG (Edge et al., 2024), a RAG framework augmented with graph-based corpus structuring; (4) LightRAG (Guo et al., 2025), a lightweight dual-level RAG approach; (5) RAPTOR (Sarthi et al., 2024), a RAG method that builds a recursive tree structure from documents for hierarchical retrieval; (6) HippoRAG (Wang et al., 2024a) and (7) HippoRAG2 (Gutiérrez et al., 2025), an advanced successor that integrates dense–sparse retrieval with recognition memory. All models employ the same QA reader (gpt-4o-mini), and EchoRAG uses the same retriever (NV-Embed-v2) to ensure a fair comparison. We evaluate performance using three metrics: Exact Match (EM), F1 score, and Retrieval Time.

#### 5.1.2 DATASETS

To comprehensively evaluate the QA capability and retrieval performance of EchoRAG, we categorize our benchmark datasets into Simple QA, which focuses on retrieving factual knowledge from a single passage using datasets like Natural Questions (NQ) (Wang et al., 2024b) and PopQA (Mallen et al., 2022), and Multi-hop QA, which evaluates reasoning ability across multiple passages using datasets like MuSiQue (Trivedi et al., 2022), 2WikiMultihopQA (Ho et al., 2020), and HotpotQA (Yang et al., 2018).

Following the setup in HippoRAG (Wang et al., 2024a), we uniformly sample 1,000 queries from each benchmark dataset, including PopQA (Mallen et al., 2022), Natural Questions (NQ) (Wang et al., 2024b), MuSiQue (Trivedi et al., 2022), 2WikiMultihopQA (Ho et al., 2020), and HotpotQA (Yang et al., 2018). The detailed statistics of these datasets, including the number of queries and passages, are summarized in Table 1.

Table 1: Dataset statistics

| Dataset | NQ | PopQA | MuSiQue | 2Wiki | HotpotQA |
|---|---|---|---|---|---|
| Number of queries | 1,000 | 1,000 | 1,000 | 1,000 | 1,000 |
| Number of passages | 9,633 | 8,676 | 11,656 | 6,119 | 9,811 |

## 5.2 RESULTS

### 5.2.1 OVERALL QA PERFORMANCE

We conduct a comprehensive comparison between EchoRAG and all baseline methods on five benchmark datasets to evaluate overall QA performance. The results are summarized in Table 2, covering accuracy metrics (EM and F1). All baselines follow the same experimental setup described above.

Table 2: EM and F1 for each retrieval method across datasets. Bold indicates EchoRAG results.

| Method | NQ | | PopQA | | MuSiQue | | 2Wiki | | HotpotQA | |
|---|---|---|---|---|---|---|---|---|---|---|
| | EM | F1 | EM | F1 | EM | F1 | EM | F1 | EM | F1 |
| None | 35.2 | 52.7 | 16.1 | 22.7 | 11.2 | 22.0 | 30.2 | 36.3 | 28.6 | 41.0 |
| NV-Embed-v2 (Lee et al., 2024) | 43.5 | 59.9 | 41.7 | 55.8 | 32.8 | 46.0 | 54.4 | 60.8 | 57.3 | 71.0 |
| GraphRAG (Edge et al., 2024) | 38.0 | 55.5 | 30.7 | 51.3 | 27.0 | 42.0 | 45.7 | 61.0 | 51.4 | 67.6 |
| LightRAG (Guo et al., 2025) | 2.8 | 15.4 | 1.9 | 14.8 | 2.0 | 9.3 | 2.5 | 12.1 | 9.9 | 20.2 |
| RAPTOR (Sarthi et al., 2024) | 37.8 | 54.5 | 41.9 | 55.1 | 27.7 | 39.2 | 39.7 | 48.4 | 50.6 | 64.7 |
| HippoRAG (Wang et al., 2024a) | 37.2 | 52.2 | 42.5 | 56.2 | 24.0 | 35.9 | 59.4 | 67.3 | 46.3 | 60.0 |
| HippoRAG2 (Gutiérrez et al., 2025) | 43.4 | 60.0 | 41.7 | 55.7 | 35.0 | 49.3 | 60.5 | 69.7 | 56.3 | 71.1 |
| **EchoRAG** | **44.4** | **60.6** | **43.1** | **56.3** | **36.5** | **50.2** | **63.5** | **72.9** | **56.6** | **72.1** |

As demonstrated in Table 2, our experimental results validate the core hypothesis that EchoRAG significantly outperforms existing state-of-the-art methods. Our model consistently achieves the highest Exact Match (EM) and F1 scores across all five challenging benchmarks. For instance, on the complex multi-hop reasoning dataset MuSiQue, EchoRAG achieves an EM score of 36.5, surpassing the strong HippoRAG2 baseline by 1.5%. This advantage is even more pronounced on 2Wiki, where EchoRAG reaches 63.5 EM, 3.0% improvement over HippoRAG2. This consistent superiority is not merely an incremental gain but direct empirical evidence of our framework's ability to overcome the "localized reasoning" limitations that plague conventional RAG systems. EchoRAG emulates human cognitive processes by constructing a multi-dimensional knowledge graph around the "semantic gist" and employing the Cognitive Diffusion Module for global, context-aware reranking. Through this process, it successfully builds a coherent semantic scene for reasoning, which substantiates the effectiveness of our proposed approach.

### 5.2.2 HYPERPARAMETER STUDY: ENTITY-FREQUENCY REWARD $\alpha, \beta$

We analyze the effect of the entity-frequency reward in the personalization vector reward$(c_e)$ in equation 11. Larger $\alpha$ strengthens the reward; larger $\beta$ accelerates saturation as $c_e$ grows. We sweep $\alpha, \beta$ and report accuracy (Recall, EM, F1). Table 3 shows five representative settings.

We validate the effectiveness of the entity-frequency reward, a key mechanism designed to operationalize the cognitive theory of "importance judgment" within our Cognitive Diffusion Module. As shown in Table 3, the setting of hyperparameters $\alpha$ and $\beta$ is critical to model performance. Disabling this reward entirely by setting $\alpha = 0.00$ serves as an ablative baseline, yielding an EM score of 0.6002 on Musique. By introducing and carefully tuning this reward to our optimal configuration ($\alpha = 2.0$, $\beta = 1.2$), the EM score signif-

Table 3: Effect of the entity-frequency reward $(\alpha, \beta)$ on Musique. Metrics reported: Recall@5/20, EM, and F1.

| $\alpha$ | $\beta$ | R@5 | R@20 | EM | F1 |
|---|---|---|---|---|---|
| 0.00 | 0.50 | 0.9022 | 0.9603 | 0.6002 | 0.6989 |
| 1.00 | 0.50 | 0.9213 | 0.9677 | 0.6142 | 0.7106 |
| 1.00 | 0.80 | 0.9257 | 0.9695 | 0.6226 | 0.7175 |
| 1.00 | 1.10 | 0.9295 | 0.9708 | 0.6266 | 0.7218 |
| **2.00** | **1.20** | **0.9353** | **0.9735** | **0.6346** | **0.7290** |

icantly increases to 0.6346, a substantial gain of 3.44%. This result provides strong quantitative evidence for our central claim: emulating importance judgment by rewarding entities that are central to the semantic context is not just beneficial but essential. It allows EchoRAG to identify and prioritize core concepts within the retrieved knowledge, thereby constructing a more coherent semantic scene and avoiding the localized reasoning pitfalls of previous methods.

### 5.2.3 ABLATION STUDY: COMPONENT ANALYSIS

To thoroughly evaluate the importance of each proposed innovation, we perform an ablation study on the Musique dataset, with results summarized in Table 4. Specifically, we consider the following variants: (1) **w/o Gist**: removing semantic gist distillation, where the knowledge graph is directly constructed from raw retrieved content; (2)**w/o CDF (Cognitive Diffusion Module)**: removing the diffusion process, so no importance propagation across entities is performed; (3) **w/o CogniRank** : removing the final reranking module, where passages are ranked only by diffusion scores without semantic–structural fusion.

The results show that EchoRAG (Full) consistently achieves the best EM and F1 while maintaining reasonable efficiency. Removing **gist** leads to both accuracy degradation and significant overhead, demonstrating that gist distillation is indispensable for compressing redundant information and ensuring high-quality graph construction. Eliminating the **Cognitive Diffusion Module** breaks the flow of global contextual signals, resulting in lower accuracy and confirming the critical role of diffusion in capturing structural importance. Finally, removing **CogniRank** causes the most severe accuracy drop, showing that relying solely on diffusion scores cannot balance semantic and structural relevance, and that our weighted fusion reranking is necessary to produce robust knowledge ordering.

Table 4: Ablation study results for EchoRAG on the Musique dataset.

| Method | Musique | | |
|---|---|---|---|
| | EM | F1 | Time(s) |
| **EchoRAG (Full)** | **36.50** | **50.23** | 1700.58 |
| w/o Gist | 35.54 | 49.22 | 2000.44 |
| w/o CDF | 35.24 | 49.27 | 1788.03 |
| w/o CogniRank | 34.40 | 47.47 | **1412.68** |

In summary, the three modules are complementary and indispensable: gist distillation ensures semantic condensation and efficient graph building, cognitive diffusion guarantees global propagation of relevance, and CogniRank achieves a deep integration of textual and structural signals at the final ranking stage. Their synergy constitutes the core strength of EchoRAG and clearly outperforms any reduced configuration.

Due to space constraints, the discussions on retrieval time and additional experiments are presented in the Appendix A.

## 6 CONCLUSION

We presented EchoRAG, a cognitively inspired RAG framework that tackles the semantic fragmentation of conventional methods by simulating human episodic memory. Through semantic gist extraction and the novel Cognitive Diffusion Module, EchoRAG constructs a holistic semantic landscape and prioritizes central concepts via the CogniRank algorithm, enabling retrieval that is both globally aware and contextually coherent.

Experiments across multiple QA benchmarks confirm that EchoRAG achieves state-of-the-art accuracy in both simple and multi-hop reasoning tasks while maintaining high retrieval efficiency. The work demonstrates the significant potential of integrating cognitive principles into RAG design, paving the way for more semantically faithful and scalable knowledge-intensive systems. Due to space constraints, the discussions on limitations and future work are presented in Appendix A.7.

## 7 ETHICS STATEMENT

The authors have adhered to the ICLR Code of Ethics. This research is based on publicly available datasets, and their use is in full compliance with their respective licenses and terms of service. This study did not involve human subjects, and no new data containing personally identifiable information was collected. The authors declare no competing interests or potential conflicts of interest. We are committed to the principles of responsible AI development and transparent research.

## 8 REPRODUCIBILITY STATEMENT

To ensure the reproducibility of our research, we have made our code, data, and experimental setup fully available. The complete source code for our proposed EchoRAG framework, along with scripts to replicate all experiments, is provided as supplementary material and will be released on GitHub upon publication. A detailed description of our model architecture and the CogniRank algorithm is presented in Section 4. The public datasets used in our evaluation are listed in Section 5.

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

# A  APPENDIX

## A.1  EVALUATION METRICS: EXACT MATCH AND F1 SCORE

To evaluate the performance of our system on question answering datasets, we adopt two widely used metrics: Exact Match (EM) and F1 score.

Exact Match (EM) measures the percentage of predictions that match any one of the ground-truth answers exactly.

F1 Score measures the overlap between the predicted and reference answer at the token level, treating the problem as a bag-of-tokens comparison.

Formally, for a predicted answer $p$ and a reference answer $r$:

$$\text{Precision} = \frac{|p \cap r|}{|p|}, \quad \text{Recall} = \frac{|p \cap r|}{|r|}$$

$$\text{EM} = \begin{cases} 1, & \text{if } p = r \\ 0, & \text{otherwise} \end{cases}$$

$$\text{F1} = \frac{2 \cdot \text{Precision} \cdot \text{Recall}}{\text{Precision} + \text{Recall}}$$

When multiple reference answers exist, the maximum EM or F1 across all references is taken.

## A.2  GIST-BASED GRAPH CONSTRUCTION ON 2WIKI

We further evaluate gist-based graph construction on the **2Wiki** dataset. Table 5 summarizes the comparison among HippoRAG2, EchoRAG without gist, and EchoRAG with gist. The results presented in Table 5 validate the effectiveness of our design through a step-by-step comparison. First, we demonstrate the value of Cognitive Diffusion Module and CogniRank by comparing the "Without Gist" variant to HippoRAG2. Our Cognitive Diffusion Module and CogniRank significantly outperform HippoRAG2 in both efficiency and accuracy, with a notable 3.1% improvement in EM. This gain is driven by both the Cognitive Diffusion Module, which creates a superior candidate pool as evidenced by higher recall at R@100 and R@200, and the CogniRank algorithm, which precisely identifies top documents, leading to a higher R@5. Building on this strong foundation, the addition of semantic gist ("With Gist") further enhances performance. It simultaneously reduces retrieval latency by an additional 13.6% and boosts QA accuracy, confirming that distilling gist is crucial for achieving optimal results in both efficiency and effectiveness.

| Method | Total Time (s) | R@5 | R@50 | R@100 | R@150 | R@200 | EM | F1 |
|---|---|---|---|---|---|---|---|---|
| HippoRAG2 | 1953.51 | 0.9062 | 0.9748 | 0.9835 | 0.9865 | 0.9892 | 0.5940 | 0.6988 |
| Without Gist | 1899.44 | 0.9305 | **0.9830** | 0.9872 | 0.9905 | 0.9915 | 0.6253 | 0.7267 |
| With Gist | **1641.26** | **0.9353** | **0.9830** | **0.9900** | **0.9922** | **0.9928** | **0.6346** | **0.7290** |

Table 5: Comparison of HippoRAG2, EchoRAG without gist, and EchoRAG with gist on 2Wiki.

## A.3  RETRIEVAL EFFICIENCY COMPARISON

Beyond accuracy, retrieval efficiency is a crucial factor for practical deployment of RAG systems. Table 6 reports the average retrieval latency across five multi-hop QA datasets.

In a comparison of retrieval efficiency, LightRAG suffers from the highest latency, exceeding 2500 seconds on most datasets. While HippoRAG2 improves efficiency, it still requires more than 2100 seconds on the MuSiQue and HotpotQA datasets. In contrast, EchoRAG consistently achieves the lowest retrieval time, reducing latency by 28.1% on NQ, 13.7% on PopQA, and over 20-30% on the other benchmarks. These results highlight that EchoRAG not only improves accuracy but also

Table 6: Retrieval latency (s) for LightRAG, HippoRAG2, and EchoRAG across datasets.

| Method | NQ | PopQA | MuSiQue | 2Wiki | HotpotQA |
|---|---|---|---|---|---|
| | Time | Time | Time | Time | Time |
| LightRAG (Guo et al., 2025) | 2555.10 | 2306.99 | 2886.61 | 2440.94 | 2828.21 |
| HippoRAG2 (Gutiérrez et al., 2025) | 2114.05 | 2032.66 | 2120.84 | 1935.51 | 2122.40 |
| **EchoRAG** | **1838.13** | **1753.14** | **1700.58** | **1641.26** | **1516.58** |

significantly accelerates retrieval. The performance gain stems from its gist-based graph construction and cognitive diffusion, which reduce redundant exploration while maintaining global relevance propagation. This balance between effectiveness and efficiency makes EchoRAG particularly suitable for real-world multi-hop QA applications where both speed and accuracy are essential.

A.4   COMPARISON: PASSAGES VS. TRIPLES AS ANSWERING CARRIER

To assess the impact of different *answering carriers*, we design a comparison focusing on the final answering stage: (1) **Passages-as-Carrier**: the selected triples are reverse-indexed back to their source passages, and the *original passage texts* are included in the QA messages for answer generation; (2) **Triples-as-Carrier**: the selected triples themselves are directly provided as the supporting evidence in QA messages, without converting back to passage texts.

We report EM and F1 on the MusiQue dataset to highlight the difference between these two carriers. Results are shown in Table 7.

Table 7: Comparison of using original passages vs. triples as the final answering carrier on MusiQue.

The results show that using **original passages** as the answering carrier yields higher EM and F1. This demonstrates that lexical fidelity and contextual details preserved in raw text are crucial for generating precise answers, while triples are concise but omit fine-grained semantic cues and contextual modifiers, limiting answer quality.

| Method | EM | F1 |
|---|---|---|
| Triples-as-Carrier | 14.90 | 23.06 |
| Passages-as-Carrier | **36.50** | **50.23** |

A.5   GIST AND MULTI-DIMENSIONAL KNOWLEDGE GRAPH

Through systematic analysis and experiments, this study reveals the importance and synergistic effects of each core component in the proposed cognition-inspired framework. The primary finding is that the inferential extraction of semantic gist serves as a crucial prerequisite for constructing a high-quality knowledge foundation. Ablation studies clearly demonstrate that bypassing this inferential step and directly building a knowledge graph based on the surface-level information of raw text will inevitably lead to semantic deviation during the retrieval phase. For instance, when responding to the query, "Which film and television works starring actor Chris Evans include newcomers in their cast?", a system lacking gist inference will likely fail. A vector-based RAG might retrieve irrelevant documents about "drumming newcomers" due to spurious semantic similarity , while a standard Knowledge Graph-indexed RAG might correctly link "Chris Evans" to the film "The Newcomers" but cannot verify if the cast members were actual newcomers, thus failing to answer the query's core intent. This retrieval bias, caused by "gist deficiency," highlights the necessity of simulating human "comprehending memory." EchoRAG, through its inferential step, grasps the true semantic gist—that "newcomers" refers to actors who were "in the early stages of their acting careers". More importantly, through carefully designed inferential steps, EchoRAG distinguishes between key facts for verbatim retention (e.g., names like 'Chris Evans', 'Paul Dano' ) and semantic cores requiring refinement (the concept of a 'newcomer actor' ), achieving a more human-like grasp of knowledge.

Based on this profound understanding of semantic gist, this study further verifies the effectiveness of constructing a Multidimensional Knowledge Graph that integrates multi-dimensional information. This graph not only stores entity-relationship triples (e.g., 'Chris Evans' - 'stars in' - 'The Newcomers' ) but, more importantly, anchors the semantic gist from which each triple is derived (e.g., 'Paul Dano newcomer' ) and the corresponding raw passage. This design allows the system, when generating

final answers, to leverage the graph's cross-document association capability while tracing back to the original context for accuracy. The case study clearly demonstrates that for complex questions, the model uses gist-guided retrieval to prioritize and integrate raw paragraphs rather than relying on discrete, context-poor triples. For example, to answer the query about Chris Evans' co-stars, the system retrieves the full contextual passage, "Paul Dano who were in the early stages of their acting careers". This practice of returning to the source text ensures the generated response is not only accurate but also contextually rich and fluent, effectively avoiding the unnatural generation and context deficiency that arise from relying solely on structured knowledge.

## A.6 COGNITIVE DIFFUSION AND GLOBAL SEMANTIC LANDSCAPE

A central innovation of EchoRAG is the Cognitive Diffusion Module, which operationalizes principles from cognitive neuroscience to simulate the cognitive processes of episodic memory construction and importance judgment. Unlike conventional methods that rely on localized, single-step retrieval or iterative multi-hop reasoning, our cognitive diffusion mechanism constructs a global "semantic landscape" by propagating relevance across the entire knowledge graph. Our experiments reveal a previously overlooked failure mode in existing methods, particularly on complex knowledge bases like 2WikiMultihopQA: their inability to effectively model the significance derived from conceptual repetition. Human memory reinforces core concepts through repeated exposure across different contexts; similarly, our cognitive diffusion process naturally captures this global importance by integrating signals from both entity frequency (a proxy for repetition) and structural relevance, enabling a more accurate knowledge ranking. As illustrated in Figure 3, while traditional retrieval perceives isolated knowledge points, our approach comprehends a holistic view of the knowledge terrain, identifying core concepts as prominent peaks within the landscape.

The implications of this unified mechanism extend beyond the immediate scope of RAG, showcasing the viability of translating established cognitive theories into concrete computational models. By simultaneously achieving contextual scene construction (simulating episodic memory) and global importance calculation (simulating importance judgment) within a single, unified diffusion process, we offer a novel perspective on bridging the gap between symbolic reasoning and distributed vector representations. This demonstrates a promising pathway for developing AI systems with more profound and human-like reasoning capabilities.

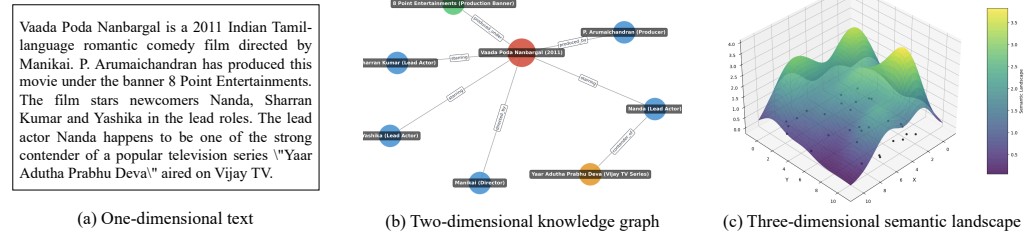

(a) One-dimensional text  (b) Two-dimensional knowledge graph  (c) Three-dimensional semantic landscape

Figure 3: This figure presents three distinct modalities of information representation: (a) a one-dimensional textual representation, (b) a two-dimensional knowledge graph, and (c) a three-dimensional semantic landscape.

## A.7 LIMITAION AND FUTURE WORK

Despite the promising results, this work has several limitations that open avenues for future research. The quality of the multi-dimensional knowledge graph is heavily dependent on the initial semantic gist extraction step, which relies on a prompted LLM. While effective, this process may occasionally produce summaries that are imprecise or fail to capture subtle but crucial nuances in the source text, potentially propagating errors into the knowledge base's foundation. Our empirical validation, while extensive, is focused on question-answering benchmarks; the framework's effectiveness on other knowledge-intensive tasks, such as summarization or conversational AI, remains to be explored.

Building upon the current framework, we identify several exciting directions for future research. To address the reliance on prompted gist extraction, one promising avenue is to develop more

robust, potentially supervised or trainable, models for gist distillation. This could ensure higher fidelity and consistency in capturing the core semantics of complex passages. To further validate the generalizability of our cognitive-inspired approach, we will adapt and evaluate EchoRAG on a broader spectrum of knowledge-intensive tasks, including long-document summarization and open-domain dialogue systems, aiming to create more holistic and context-aware AI systems.

## B  LLM USAGE

Large Language Models (LLMs) were used solely to enhance the linguistic quality of the manuscript, aiding in refining language, improving readability, ensuring clarity, rephrasing sentences, checking grammar, and boosting text flow, without any involvement in scientific content or data analysis. The authors fully take responsibility for the manuscript's content, and have ensured all content follows ethical guidelines and avoids plagiarism or scientific misconduct.

