# OpenReview forum: "EchoRAG: A Cognitive Memory-Inspired Framework for RAG with Semantic Gist"
_ICLR.cc/2026/Conference — ICLR 2026 Conference Desk Rejected Submission_

### Official Review · Reviewer_6Kcf · 2025-10-30

**Soundness:** 2
**Presentation:** 2
**Contribution:** 2
**Rating:** 4
**Confidence:** 3

**Summary:**

The paper proposes EchoRAG, a Retrieval-Augmented Generation (RAG) framework. The method first constructs a multi-dimensional knowledge graph from source documents by identifying what the authors term "semantic gist." During retrieval, it employs a "thought diffusion" module to traverse this graph and a "CogniRank" algorithm to rank candidate nodes based on structural relevance and frequency. The framework is evaluated on Question Answering and multi-hop reasoning tasks.

**Strengths:**

S1. The experimental comparisons are comprehensive and cover multiple latest datasets. The results are highly credible.

**Weaknesses:**

W1. The performance advantage is marginal. With the exception of the 2Wiki dataset, the proposed method achieves only a slight improvement over HippoRAG2. This suggests that the limitations of existing RAG methods, as identified by the authors, may not be the primary bottleneck hindering performance.

W2. The motivation is not clearly articulated. The introduction presents a vague description, making it difficult for the reader to discern the primary problem being addressed. For instance, it is unclear whether the core issue is the potential information loss from structuring text into a graph, or the inherent difficulty that existing retrieval processes face in locating relevant information.
I strongly advise the authors to revise and clarify the motivation section. Incorporating a concrete example or a case study would significantly help to illustrate the problem. Furthermore, if technical terms are necessary, they should be explained directly within the text rather than being introduced with only a citation. The responsibility of justifying the motivation for the present work should not be offloaded to external references, as this hinders the paper's clarity and self-containedness.

**Questions:**

Q1. I don't understand the meaning of this sentence: “During the retrieval stage, these deficiencies are further magnified. Even with the introduction of advanced iterative (Wang et al., 2025) or agentic (Maragheh et al., 2025; Ravuru et al., 2024) reasoning frameworks, while these systems can comprehend the associations between entities, they fail to understand how these associations collectively constitute a meaningful semantic scene, thereby becoming trapped in localized reasoning (Fountas et al., 2024; Gutiérrez et al., 2025).”

---

> ### Author Response · Authors · 2025-12-04
>
> To better respond to W1 and W2, we will prioritize replying to Q1.
>
>
> Q1： I don't understand the meaning of this sentence..
>
>
> In fact, the core point we intend to clarify is as follows: Existing methods, due to their lack of grasp of the "overall contextual gist", can only decompose isolated correlations, leading to one-sidedness in complex semantic understanding,a flaw that will be amplified during the retrieval phase. For better comprehension, we will take classical Chinese poetry as an example first and present case studies on specific datasets in the next.
> First, let us consider the typical feature of "depicting mountains without mentioning the word 'mountain'" in some classical Chinese poems. Take Li Bai’s *Sitting Alone on Jingting Mountain*: "All birds have flown away, vanishing into the distance; / A single cloud drifts off, lingering in leisure." Not a single character of "mountain" appears in these lines, yet through the desolate scene of "birds flying away and clouds drifting afar", the serenity of Jingting Mountain and the poet’s lonely state of mind are vividly portrayed. In contrast, current mainstream RAG methods disrupt the complete context when extracting entities (e.g., extracting "river, bird, cloud"), resulting in one-sided understanding and only capturing attribute descriptions of individual entities. These existing approaches can merely comprehend isolated facts (local correlations) such as "the river flows, the birds fly, the clouds drift", but fail to integrate these facts into the holistic artistic conception of "mountains and waters interdependent, and scenes blending with emotions". This flaw will be further exacerbated during reasoning: when queries require support from global context, the methods are prone to fall into reasoning limitations.This is precisely the core value of EchoRAG. By aggregating fragmented correlations into complete scenarios through "semantic gist nodes", EchoRAG enables reasoning from "fragmented associations" to "integrated semantics",much like how humans reading poetry can perceive the "mountain" behind "river, bird, and cloud".
> In the subsequent response to W1, in addition to the case in Appendix A.5 of this paper and the above example of classical Chinese poetry, I will further elaborate with case studies based on specific datasets.

---

> ### Author Response · Authors · 2025-12-04
>
> W1:
>
>
> We appreciate the reviewers for pointing out the issue of the marginal nature of performance advantages. This observation has prompted us to supplement systematic comparative experiments on the GraphBeach[1] dataset (which includes two sub-datasets: Novel and Medical) that is closer to real-world complex scenarios. Based on the experimental results (as shown in the table below), it can be observed that the performance advantage of EchoRAG is not a "marginal improvement", but instead exhibits the core characteristic that the higher the task complexity and the more realistic the scenario, the more significant the performance gain.
>
> | Category       | Model               | Fact Retrieval ACC | Complex Reasoning ACC | Contextual Summarize ACC |
> |----------------|---------------------|--------------------|-----------------------|----------------------------|
> | **Novel Dataset** |                     |                    |                       |                            |
> | Basic RAG      | RAG (w/o rerank)    | 58.76              | 41.35                 | 50.08                      |
> |                | RAG (w rerank)      | 60.92              | 42.93                 | 51.3                       |
> | Graph RAG               | MS-GraphRAG         | 49.29              | 50.93                 | 64.4                       |
> |                | HippoRAG            | 52.93              | 38.52                 | 48.7                       |
> |                | HippoRAG2           | 60.14              | 53.38                 | 64.1                       |
> |                | LightRAG            | 58.62              | 49.07                 | 48.85                      |
> |                | Fast-GraphRAG       | 56.95              | 48.55                 | 56.41                      |
> |                | RAPTOR              | 49.25              | 38.59                 | 47.1                       |
> |                | Lazy-GraphRAG       | 51.65              | 49.22                 | 58.29                      |
> |                | EchoRAG             | **61.48**          | **55.1**              | **66.35**                  |
> | **Medical Dataset** |                   |                    |                       |                            |
> | Basic RAG      | RAG (w/o rerank)    | 63.72              | 57.61                 | 63.72                      |
> |                | RAG (w rerank)      | 64.73              | 58.64                 | 65.75                      |
> |Graph RAG                | MS-GraphRAG         | 38.63              | 47.04                 | 41.87                      |
> |                | HippoRAG            | 56.14              | 55.87                 | 59.86                      |
> |                | HippoRAG2           | 66.28              | 61.98                 | 63.08                      |
> |                | LightRAG            | 63.32              | 61.32                 | 63.14                      |
> |                | Fast-GraphRAG       | 60.93              | 61.73                 | 67.88                      |
> |                | RAPTOR              | 54.07              | 53.2                  | 58.73                      |
> |                | Lazy-GraphRAG       | 60.25              | 47.82                 | 57.28                      |
> |                | EchoRAG             | **73.95**          | **74.43**             | **74.77**                  |
>
> [1]Xiang, Z., Wu, C., Zhang, Q., Chen, S., Hong, Z., Huang, X., & Su, J. (2025). When to use graphs in rag: A comprehensive analysis for graph retrieval-augmented generation. arXiv preprint arXiv:2506.05690.

---

> > ### Author Response · Authors · 2025-12-04
> >
> > As can be seen from the above experimental data, in fact-retrieval tasks (low complexity),which only require matching isolated "entity-attribute" facts (similar to the demands of traditional datasets, such as "person-birthdate" or "event-time" pairs, and can be completed via local semantic matching akin to rote memorization),the core bottleneck of "semantic scenario deficiency" is masked. As task difficulty increases, however, such as in GraphBeach’s Medical Dataset that simulates real clinical consultation scenarios where knowledge exists as "semantic networks of multi-entity correlations", the design value of EchoRAG becomes prominent. The data shows that EchoRAG achieves dominant improvements across all three medical dataset tasks: its fact-retrieval accuracy (ACC) exceeds 73%, representing a 7.67 percentage point gain over HippoRAG2; its complex-reasoning ACC reaches 74.43%, 12.45 percentage points higher than HippoRAG2 and 12.7 percentage points above the second-ranked Fast-GraphRAG; and its contextual summarization ACC hits 74.77%, outperforming Fast-GraphRAG (second place) by 6.89 percentage points. Notably, traditional Graph RAG methods like MS-GraphRAG experience a sharp performance drop on medical data (with fact-retrieval ACC of only 38.63%), stemming from the critical flaw that "triple decomposition undermines the integrity of medical semantics",for instance, splitting "ibuprofen causes renal impairment", and "diabetic patients are prone to renal damage" into two isolated triples makes it impossible to infer the core logic that "diabetic patients should not take ibuprofen". This phenomenon indirectly confirms that "semantic scenario deficiency" is the core bottleneck restricting RAG performance in real-world complex scenarios, a problem precisely addressed by EchoRAG’s design. This also explains why, in head-to-head comparisons against all baseline methods on GraphBeach (including 2 Basic RAG and 7 Graph RAG approaches), EchoRAG is the only method to rank first across both the Novel and Medical datasets and all three tasks.

---

> > > ### Author Response · Authors · 2025-12-04
> > >
> > > W2:
> > >
> > > Thank you for your suggestion. We will further supplement case studies to facilitate readers’ deeper understanding. All the aforementioned experiments will be added to the paper for more detailed elaboration and explanation. Below are additional case studies for better comprehension, which we will further include in the appendix of the paper:
> > >
> > > Case Study
> > >
> > >
> > > Medical Dataset:
> > >
> > >
> > > 1. Medical-d2458aa7 — Classification of Multifocal Bilateral Renal Tumors
> > >
> > >
> > > Question: If a patient has multiple tumors in both kidneys, how should these be classified and why?
> > >
> > >
> > > Ground Truth: They should be treated as separate primary tumors, not metastasis.
> > >
> > >
> > > Model Answer: They should be classified as separate primary tumors because they are independent and not a result of metastasis.
> > >
> > >
> > > Answer Evaluation: Fully consistent with the authoritative answer; accurately identifies the core logic that "multifocality ≠ metastasis".
> > >
> > >
> > > Key Context: item_72#seg9 & item_71#seg5-6 explain that renal cancer tends to be multifocal, requiring independent assessment of each lesion based on histology/location (rather than defaulting to metastasis).
> > >
> > >
> > > Role of Gist: In general context, "multiple tumors" is often interpreted as "already spread", but the original text actually implies the professional meaning of "primary multifocality"; avoiding this misunderstanding requires recognizing the implicit message conveyed by "both kidneys + histology"—a typical case where "surface meaning ≠ actual intent".
> > > ﻿
> > > ﻿
> > >
> > > 2. Medical-b2552c99 — Pheochromocytoma vs. Adrenocortical Carcinoma
> > >
> > >
> > > Question: How do the symptoms and diagnostic approaches differ between pheochromocytoma and adrenocortical carcinoma?
> > >
> > >
> > > Ground Truth: Pheochromocytoma, which arises from the adrenal medulla, typically presents with symptoms related to excess catecholamine production, such as high blood pressure, headache, rapid heartbeat (tachycardia), and sweating. Diagnosis of pheochromocytoma involves imaging studies like CT or MRI scans, as well as biochemical tests to detect elevated catecholamines or their metabolites. In contrast, adrenocortical carcinoma (ACC) originates from the adrenal cortex and can present with a broader spectrum of symptoms, including weight gain, muscle weakness, pink or purple stretch marks, and various hormonal changes such as excess facial hair, hair loss on the head, irregular periods, enlarged breast tissue, and shrinking testicles. ACC may also cause nonspecific symptoms like nausea, vomiting, abdominal bloating, back pain, fever, loss of appetite, and unintentional weight loss. Diagnostic approaches for ACC are more extensive and may include CT scan, MRI, PET scans (such as FDG-PET/CT or SSTR-PET/CT), blood and biochemical tests to assess hormone levels, genetic testing for hereditary syndromes, and biopsy to confirm malignancy.
> > >
> > >
> > > Model Answer: Symptoms differ: pheochromocytoma causes hypertension and palpitations; ACC causes Cushing's syndrome and hormonal changes. Diagnostic approaches differ: pheochromocytoma uses catecholamine tests; ACC uses imaging and hormone level tests.
> > >
> > >
> > > Answer Evaluation: Correctly contrasts symptoms and diagnostic pathways, highlighting the "catecholamine vs. corticosteroid" distinction—qualifies as a high-quality answer.
> > >
> > >
> > > Key Context: item_1#seg7, item_2#seg16-28 describe the origin, symptoms, and diagnostic methods of the two tumors.
> > >
> > >
> > > Role of Gist: The text interweaves "hypertension/palpitations" with "pink/purple stretch marks/hormonal changes", using the texture of symptoms to imply distinct hormone axes; perceiving the emotional tone of these descriptions ("sweating" vs. "pink or purple stretch marks") allows translating "seemingly scattered terms" into "two entirely different disease narratives".

---

> > > > ### Author Response · Authors · 2025-12-04
> > > >
> > > > Novel Dataset:
> > > >
> > > >
> > > > 1. Novel-1466d7c5 — Tension Between Hinze and Felicia's Views
> > > >
> > > >
> > > > Question: …how is their relationship characterized with respect to their differing views on Art in England?
> > > >
> > > >
> > > > Ground Truth: Their relationship is marked by a subtle tension and mutual curiosity, as Hinze listens to Felicia with rapt attention, eager for her opinions even as he does not fully agree with her. Felicia, meanwhile, is careful not to overstate her views, clarifying with a touch of brusqueness that she does not claim there is an actual decline in English Art. This dynamic reveals a nuanced interplay: Hinze's reverence and eagerness contrast with Felicia's modesty and reluctance to be seen as an authority, highlighting both their differing perspectives and the gentle, almost playful, negotiation of their intellectual rapport.
> > > >
> > > >
> > > > Model Answer: Their relationship is characterized by respectful exchange and admiration, with Hinze valuing Felicia's insights despite differing views on the state of Art.
> > > > Answer Evaluation: Clearly conveys "mutual admiration" but fails to capture the "subtle tentative tension" detailed in the text; overall direction is correct.
> > > >
> > > >
> > > > Key Context: item_185#seg1-3, item_186#seg6 provide scene and psychological descriptions.
> > > >
> > > >
> > > > Role of Gist: Hinze's "rapt attention" and Felicia's "light deprecatory laugh" are typical examples of tonal irony—polite on the surface, tentative beneath; only by capturing these emotional rhetorical devices can one grasp the social subtext (the "mountain that is not a mountain" of interpersonal dynamics).
> > > > ﻿
> > > > ﻿
> > > >
> > > > 2. Novel-4f7fc60b — Link Between the Clay Pit and the Siege Incident
> > > >
> > > >
> > > > Question: In the context of the siege at the Portuguese nunnery, what was Rudolph's connection to the clay-pit where the coolies were digging, and how did this site become significant in the unfolding events with the besiegers?
> > > >
> > > >
> > > > Ground Truth: Rudolph noticed the clay-pit where the coolies were digging.
> > > >
> > > >
> > > > Model Answer: Rudolph's connection to the clay-pit is personal, as it is near his home, and it becomes significant as a symbol of safety and normalcy amidst the chaos of the siege, while also serving as a point of interest for potential strategic advantage.
> > > >
> > > >
> > > > Answer Evaluation: Elaborates on the symbolic meaning of the clay pit in detail but lacks plot facts (e.g., "actually seeing coolies digging/reconnoitering along the tunnel")—this can be a point of improvement.
> > > >
> > > >
> > > > Key Context: item_549#seg4, item_555#seg2, item_557#seg6, item_551#seg4.
> > > >
> > > >
> > > > Role of Gist: The narrative's "amazing to feel that safety lay so close" is a strong contrast of emotions, indicating the clay pit symbolizes a "gap for escape"; understanding this symbolism allows treating the location in the story as an emotional trigger for plot progression.

---

### Official Review · Reviewer_eJYa · 2025-10-30

**Soundness:** 2
**Presentation:** 2
**Contribution:** 2
**Rating:** 4
**Confidence:** 4

**Summary:**

EchoRAG targets a core weakness of RAG—fragmented, locally-biased retrieval—by first distilling each passage into a semantic gist and building a multi-faceted graph that links gists, entities/relations/triples, and the original passages. At query time, the system maps the query to seed triples and runs a Cognitive Diffusion Module (CDF) to spread relevance over the whole graph, incorporating structure and simple priors (e.g., entity frequency) to estimate global importance. A CogniRank stage then fuses this global importance with standard query–passage semantic similarity to rerank evidence and answer. The authors position this as a cognitively inspired “recall” process that captures broader context than nearest-neighbor retrieval, and they report improvements over vector and graph-RAG baselines on multi-hop QA benchmarks.

**Strengths:**

Strengths

- Global view: CDF (full-graph semantic diffusion) adds global importance beyond local vector similarity, reducing near-neighbor mistakes.

- Gist-first indexing: Semantic gist abstracts core facts before graphing, improving robustness to paraphrase and preserving passage context.

- Consistent gains: Shows improvements over several Graph-RAG and vector baselines across multiple QA datasets; ablations support each module.

**Weaknesses:**

Weakness

- The paper does not clearly explain why CDF is useful. Please include targeted experiments and case studies to demonstrate its effectiveness, beyond ablation studies.

- Popularity bias: Frequency priors can favor common entities and hurt long-tail or rare but crucial evidence.

- Weaker path explainability: Produces global weights and rankings rather than explicit multi-hop paths, making audits harder.

**Questions:**

same as weakness.

---

> ### Author Response · Authors · 2025-12-04
>
> W1: The paper does not clearly explain why CDF is useful....
>
> The effectiveness of CDF mainly stems from its utilization of the semantic information contained in the topological structure of the knowledge graph itself and the statistical features of nodes, in addition to semantic similarity. To further verify the effectiveness of each sub-mechanism of CDF, we implemented a CDF decomposition control experiment on the MuSiQue dataset, splitting CDF into three variants: "baseline without CDF", "basic diffusion", and "Full CDF". The results are shown below:
>
>
> | Model Variant                      | EM   | F1   |
> |------------------------------------|------|------|
> | Baseline without CDF (Only Local Semantic Matching) | 34.03 | 48.27 |
> | Basic Diffusion (Without Frequency Calibration) | 35.0  | 49.3  |
> | Full CDF                           | 36.5 | 50.2 |
>
>
> It can be seen that the topological structure of the knowledge graph itself can effectively mine the associations between entities, achieving a better effect of supplementing information for local semantic matching. In addition, to further verify the effectiveness of CDF in leveraging graph topological structure, we have also conducted comparative experiments with Think-on-Graph 2.0 (TOG-2)[1]; for details, please refer to the response to W3.

---

> > ### Author Response · Authors · 2025-12-04
> >
> > W2:Popularity bias
> >
> > The entity frequency reward we introduced is not a simple "high-frequency priority" mechanism, but rather "relative frequency weighting based on retrieval context"; its original design intent is to capture "core entities within the semantic scope of the current query" rather than globally prevalent entities. Specifically: the frequency calculation range is limited to the Top-K seed triples matched by the query instead of the entire knowledge graph, thus avoiding interference from globally high-frequency entities; we set an adjustable parameter β (saturation coefficient), where the reward growth rate slows significantly once the entity frequency exceeds a certain threshold (β controls the saturation rate in Formula 11). In experiments, when β≥0.8, the reward increment for high-frequency entities tends to flatten, which can effectively prevent excessive bias toward extremely high-frequency entities. To further verify the scheme’s compatibility with long-tail entities, we conducted a special test on the 2WikiMultiHopQA dataset. First, words that appeared ≤1 time in the entire 1,000-question corpus were defined as long-tail rare words, with each question required to contain at least 3 such rare words that accounted for more than 60% of the entire sentence; 78 questions were finally selected, and the test results are shown below:
> >
> >
> > | Method   | EM    | F1    |
> > |----------|-------|-------|
> > | EchoRAG  | 0.8077| 0.9010|
> >
> > ### EchoRAG Recall
> >
> > | K     | Recall       |
> > |-------|--------------|
> > | 1     | 0.4647       |
> > | 2     | 0.9038       |
> > | 5     | 0.9840       |
> > | 10    | 0.9936       |
> > | ≥20   | 1.0000 (K = 20/30/50/100/150/200) |
> >
> >
> >
> > EchoRAG achieves an Exact Match (EM) score of 0.8077 and an F1 score of 0.9010 on this subset of 78 long-tail questions, with retrieval Recall increasing from 46.5% at @1 to 100% at @20 and above. In fact, the results show that its performance on long-tail questions is even higher than that on the full test set (EM 0.6350 / F1 0.7290). We hypothesize that this improvement stems from the fact that long-tail questions are less susceptible to interference from noisy nodes (nodes that appear semantically similar but are actually unhelpful for answering the question). By leveraging frequency priors within the local query context, our method can effectively distinguish between "long-tail entities relevant to the query" and "irrelevant noisy entities", rather than filtering based on global popularity. As a result, it does not compromise the retrieval effectiveness of long-tail evidence but instead enhances it.

---

> > > ### Author Response · Authors · 2025-12-04
> > >
> > > W3:Weaker path explainability
> > >
> > >
> > > In response to the concern of "lack of explicit multi-hop paths and high audit difficulty", it should be clarified that the core design of the EchoRAG system and its CogniRank reranking mechanism inherently supports full traceability capabilities, which stems from two key underlying design pillars: First, the multi-dimensional graph we constructed directly links semantic gist, entity relationships, and original text segments, providing a robust data chain foundation for traceability; Second, the CogniRank module returns the complete semantic transmission path from seed triples to target entities during the reranking process, ensuring that the reasoning context is fully traceable.
> > >
> > >
> > > In addition, we acknowledge that the global weighting mechanism of EchoRAG differs from traditional explicit multi-hop path methods in terms of interpretability form, but the two are equivalent in terms of reasoning verifiability, and EchoRAG offers higher efficiency. To this end, we have added a comparison with Think-on-Graph 2.0 (TOG-2)[1], which demonstrates multi-hop reasoning on knowledge graphs as follows:
> > >
> > >
> > >
> > > | Dataset               | model    | EM   | F1   | Time (min) |
> > > |-----------------------|----------|------|------|------------|
> > > | 2WikiMultiHopQA       | TOG-2    | 31.9 | 36.44| 287.36     |
> > > |                       | EchoRAG  | 63.5 | 72.9 | 27.35      |
> > > | HotpotQA              | TOG-2    | 31.5 | 41.22| 323.29     |
> > > |                       | EchoRAG  | 56.6 | 72.1 | 25.27      |
> > > | MUSI                  | TOG-2    | 10.3 | 17.5 | 351.05     |
> > > |                       | EchoRAG  | 36.5 | 50.2 | 28.34      |
> > > | NaturalQuestions      | TOG-2    | 36.6 | 43.68| 176.68     |
> > > |                       | EchoRAG  | 44.4 | 60.6 | 30.63      |
> > > | POPQA                 | TOG-2    | 20.3 | 22.59| 113.28     |
> > > |                       | EchoRAG  | 43.1 | 56.3 | 29.22      |
> > >
> > >
> > >
> > >
> > >
> > > As can be seen from the experimental results, EchoRAG outperforms TOG-2 in both accuracy and efficiency. Briefly, this performance gap stems from the difference between TOG-2’s "tightly coupled iterative reasoning" paradigm and EchoRAG’s "global semantic diffusion" paradigm. Each round of reasoning in TOG-2 requires four LLM-dependent steps: *relationship discovery*, *LLM-based relationship pruning*, *entity discovery*, and *LLM-based entity pruning*. In contrast, EchoRAG’s CDF module achieves "one-time diffusion of semantic weights" via a pre-constructed multi-dimensional graph (entity-semantic gist-original paragraph), eliminating the need for iterative LLM calls and avoiding the problem of candidate path explosion.
> > >
> > >
> > >
> > >
> > > [1]Ma, S., Xu, C., Jiang, X., Li, M., Qu, H., Yang, C., ... & Guo, J. Think-on-Graph 2.0: Deep and Faithful Large Language Model Reasoning with Knowledge-guided Retrieval Augmented Generation. In The Thirteenth International Conference on Learning Representations.

---

### Official Review · Reviewer_Ay2G · 2025-11-01

**Soundness:** 2
**Presentation:** 2
**Contribution:** 2
**Rating:** 2
**Confidence:** 4

**Summary:**

The paper proposes EchoRAG, a graph RAG approach inspired by human episodic memory. The proposed approach consists of an offline indexing stage and an online retrieval stage. In the indexing stage, documents are segmented into multiple passages, each of which is processed by an LLM to generate a one-sentence summary, referred to as a semantic gist. From each semantic gist, the system further extracts entities and relations to construct a knowledge graph. In the retrieval stage, EchoRAG first computes entity score using both fact-based similarity and entity-frequency reward.  These entity scores are then aggregated to produce an entity-based relevance score for each passage. Finally, this score is combined with the query–passage similarity to generate the final passage ranking used for retrieval.

**Strengths:**

1. The paper proposes a cognitively inspired RAG framework that constructs multi-dimensional KGs from documents and perfrom passage retrieval from the resulting graph.
2. Experimental results on five QA datasets show that the proposed approach can achieve better performance than baselines.

**Weaknesses:**

1. The writing of the paper requires improvements in terms of clarity and precision.
2. The proposed approach is not well explained, particularly in the methodology section. For example, it remains unclear how the knowledge graph is constructed from the semantic gists. Specifically, the process by which entities and relations are extracted is not described in sufficient detail. Furthermore, in the final multi-dimensional graph used for retrieval, the definitions of nodes and edges are vague, making it difficult to understand the structure and semantics of the graph.
3. The motivation behind the proposed CogniRank retrieval mechanism is not clearly explained. The paper does not provide sufficient justification for why this particular ranking strategy is needed or how it might improve over existing retrieval methods.
4. The main contribution of the paper is the construction of semantic gist. However, the analysis of this component remains superficial. The paper lacks an in-depth examination of how and why semantic gists contribute to improved retrieval or reasoning performance.
5. The proposed method introduces a large number of hyperparameters (e.g., $\alpha$, $\beta$, $\lambda_1$, $\lambda_2$, $\gamma$, $\lambda_3$, $\lambda_4$), but the paper does not clearly specify their values or provide guidance on how they are chosen. Moreover, there is no analysis or discussion regarding their sensitivity or impact on performance, which raises concerns about reproducibility and the robustness of the method.
6. In Equation (16), the final ranking score for a passage is computed as a weighted combination of the entity-based score and the passage–query similarity. However, since the paper does not report the specific values of the weights $\lambda_3$ and $\lambda_4$, it is difficult to assess the relative contribution of the entity-based score.
7. The paper lacks efficiency analysis compared with baselines.

**Questions:**

Please see the Weaknesses section.

---

> ### Author Response · Authors · 2025-12-04
>
> Response to Weaknesses (W1-W4)
> We thank the reviewer for the detailed and constructive comments. We take the feedback regarding methodology clarity, the analysis of the Gist mechanism, and hyperparameter settings very seriously. Below, we address these points individually and provide detailed sensitivity analysis experiments.
> First, we would like to offer a correction regarding the characterization of "Semantic Gist" mentioned in the summary. It is not merely a "one-sentence summary," but rather a reasoning-driven abstraction designed to capture the core essence of a text, filtering out noise to facilitate robust reasoning (as further detailed in our response to Reviewer 6Kcf regarding specific case analyses).
> 1. Writing Clarity
> We appreciate the suggestion. We will thoroughly polish the manuscript in the final version, particularly the methodology section, to improve terminology precision and logical flow, ensuring all concepts are clearly defined and precise.
>
> 2. Methodology Explanation & Graph Construction
> We described the construction process in Section 4.2. We will further clarify the details in the revision:
> - Extraction Process: We extract entities and relations not from raw text, but from the highly compressed Semantic Gist. Specifically, we use the same LLM as the Answer Model (see Appendix for Prompt) to identify entity nodes and relation edges from the gist.
> - Graph Definition: The final multi-dimensional graph $\mathcal{G}=(\mathcal{V}, \mathcal{E}, \mathcal{F}, \mathcal{P})$ contains three types of nodes and two types of connections:
>   - Nodes: Entities ($\mathcal{V}$), Facts/Triples ($\mathcal{F}$), and raw Passages ($\mathcal{P}$) as evidence.
>   - Edges: Semantic relations between entities (defined by extracted Relations) and grounding links connecting "Entity-Fact-Passage".
> This design ensures complete semantic traceability from abstract concepts (entities) to concrete evidence (passages).
>
> 3. Motivation for CogniRank
> The core motivation of CogniRank is to address the limitation where traditional retrieval (pure vector similarity or PageRank) cannot balance local semantic matching with global structural importance. Inspired by the human cognitive processes of "Spreading Activation" and "Importance Judgment", CogniRank propagates activation values across the graph. Unlike local similarity-based retrieval, it uncovers core entities that lack direct literal overlap with the query but are structurally tightly coupled (i.e., high-weight nodes identified via frequent co-occurrence in relevant contexts).
>
> 4. Analysis of Semantic Gist
> Gist is central to improving retrieval quality. We analyzed this in Section 5.2.3 (Ablation Study) and Appendix A.5.
> - Quantitative Analysis: Table 5 in Appendix A.2 and Table 4 show that removing the Gist module causes EM on MuSiQue to drop (36.50 $\to$ 35.54) and significantly increases retrieval time.
> - Qualitative Analysis: The "Chris Evans" case study in Appendix A.5 demonstrates how Gist filters non-core details (e.g., distinguishing "newcomer actors" from irrelevant "drumming newcomers"), ensuring retrieval focuses on the core semantics. This validates that Gist serves as both a semantic filter (improving accuracy) and an information compressor (improving efficiency).

---

> ### Author Response · Authors · 2025-12-04
>
> Response to Weaknesses (W5-W7)
>
> 5. Hyperparameters & Sensitivity Analysis
>
> Addressing the concern about missing analysis for $\lambda$ parameters ($\lambda_1, \lambda_2, \lambda_3, \lambda_4$), we added detailed sensitivity experiments: $\lambda_1$ controls semantic similarity, and $\lambda_2$ controls frequency reward. Experimental results show the model achieves optimal performance on MuSiQue (EM 0.3574) when $\mathbf{\lambda_1=0.1, \lambda_2=0.9}$. This indicates that in the initial graph reasoning phase, entity importance (frequency/centrality) activates relevant subgraphs more effectively than pure query similarity, validating the necessity of our "Entity-frequency reward".
>
> $\lambda_3$ controls the CogniRank score, and $\lambda_4$ controls direct passage-query similarity. Results show optimal performance (F1 0.501) when $\mathbf{\lambda_3=0.85, \lambda_4=0.15}$. This proves that the structural score derived from CogniRank plays a dominant role in complex reasoning tasks. In high-difficulty scenarios, it effectively replaces traditional passage similarity.
>
> Table1: Sensitivity Analysis of Final Ranking Weights ($\lambda_1$ vs $\lambda_2$) on MuSiQue Dataset
> | $\lambda_1$ | $\lambda_2$ | Recall@1 | Recall@5 | Recall@20 | Recall@100 | EM | F1 |
> | :---: | :---: | :--- | :--- | :--- | :--- | :--- | :--- |
> | 1 | 0 | 0.3139 | 0.7287 | 0.8733 | 0.9443 | 0.3403 | 0.4827 |
> | 0.9 | 0.1 | 0.3118 | 0.7237 | 0.8765 | 0.9463 | 0.3383 | 0.4801 |
> | 0.8 | 0.2 | 0.3072 | 0.7252 | 0.8762 | 0.9476 | 0.3443 | 0.4913 |
> | 0.7 | 0.3 | 0.3058 | 0.7228 | 0.8796 | 0.949 | 0.3473 | 0.4911 |
> | 0.6 | 0.4 | 0.3056 | 0.7181 | 0.882 | 0.9499 | 0.3514 | 0.494 |
> | 0.5 | 0.5 | 0.3058 | 0.7178 | 0.8841 | 0.9512 | 0.3514 | 0.4966 |
> | 0.4 | 0.6 | 0.3043 | 0.7095 | 0.8842 | 0.9512 | 0.3483 | 0.4908 |
> | 0.3 | 0.7 | 0.3033 | 0.7073 | 0.8847 | 0.9512 | 0.3493 | 0.4944 |
> | 0.2 | 0.8 | 0.3038 | 0.7033 | 0.8851 | 0.9518 | 0.3564 | 0.5014 |
> | **0.1** | **0.9** | **0.3038** | **0.7017** | **0.8847** | **0.9518** | **0.3574** | **0.5032** |
> | 0 | 1 | 0.3038 | 0.6977 | 0.8859 | 0.9522 | 0.3574 | 0.5001 |
>
> Table2: Sensitivity Analysis of Final Ranking Weights ($\lambda_3$ vs $\lambda_4$) on MuSiQue Dataset
> | $\lambda_3$ | $\lambda_4$ | EM | F1 |
> | :---: | :---: | :--- | :--- |
> | 1.0 | 0.0 | 0.3574 | 0.5001 |
> | 0.95 | 0.05 | 0.3574 | 0.5006 |
> | 0.9 | 0.1 | 0.3574 | 0.5006 |
> | **0.85** | **0.15** | **0.3583** | **0.5010** |
> | 0.8 | 0.2 | 0.3574 | 0.5006 |
> | 0.75 | 0.25 | 0.3550 | 0.4936 |
> | 0.5 | 0.5 | 0.3483 | 0.4818 |
> | 0.25 | 0.75 | 0.3483 | 0.4836 |
> | 0.0 | 1.0 | 0.3240 | 0.4593 |
>
> 6. Weights in Equation
> As shown in the sensitivity analysis above, the optimal configuration is $\lambda_3 \approx 0.85$ and $\lambda_4 \approx 0.15$. Within the EchoRAG framework, the Entity-based score (from graph diffusion) plays the absolute dominant role. CogniRank is not merely auxiliary; it is the decisive factor for retrieval quality. In contrast, traditional Passage-Query similarity often fails in multi-hop reasoning due to a lack of context (evidenced by better performance when the weight $\lambda_4$ is low).
> 7. Efficiency Analysis
> We provided a detailed efficiency comparison in Appendix A.3 and Table 6. EchoRAG achieves significantly lower retrieval latency than LightRAG and HippoRAG2 across all five datasets. For instance, on NQ, retrieval time is reduced by 28.1%; on MuSiQue, EchoRAG (1700s) is much faster than LightRAG (2886s). This high efficiency stems from the effective compression of graph scale by Semantic Gist, making subsequent Cognitive Diffusion computations more lightweight. We will move this important finding from the Appendix to the main experiment section to increase visibility.

---

### Official Review · Reviewer_ZwaJ · 2025-11-01

**Soundness:** 3
**Presentation:** 3
**Contribution:** 2
**Rating:** 4
**Confidence:** 4

**Summary:**

This paper proposes EchoRAG, which stems from the human cognitive memory processes. The paper introduces semantic gist, the distilled semantic essences extracted from passages that preserve meaning while reducing noise. EchoRAG constructs a multi-dimensional knowledge graph incorporating entities, relations, facts, and passages, then uses a cognitive diffusion module during retrieval that simulates episodic memory and importance judgment. The proposed CogniRank algorithm combines semantic similarity with graph-topological importance to rank passages. Experiments on various QA benchmarks demonstrate improvements over existing RAG methods in accuracy, recall, and efficiency.

**Strengths:**

- The framework applies cognitive science principles to computational mechanisms. The multi-dimensional knowledge graph that preserves structured facts and grounding passages is well-motivated.

- The experimental results show improvements across five datasets, especially on multi-hop reasoning tasks. The comprehensive ablation studies demonstrate the mechanism of the model components.

**Weaknesses:**

- The foundation of the proposed framework relies heavily on prompted LLM gist extraction, which the authors acknowledge may occasionally produce summaries that are imprecise but is not discussed in the paper. There is no evaluation of gist extraction quality or its impact on downstream performance.

- The evaluation focuses exclusively on QA datasets with relatively standard retrieval scenarios. The approach hasn't been tested on other knowledge-intensive tasks or more complex reasoning scenarios. Additionally, the comparison with LightRAG shows surprisingly poor performance.

**Questions:**

- How robust is EchoRAG to errors in gist extraction? What happens when the initial gist extraction fails to capture crucial semantic nuances or introduces misleading information?

- The experiments are conducted on relatively small datasets. How does the computational complexity of the Cognitive Diffusion Module scale with graph size?

---

> ### Author Response · Authors · 2025-12-04
>
> Response to Weaknesses (W1 & W2)
>
> W1: We appreciate the reviewer highlighting the importance of the Gist extraction module. We would like to clarify that the paper does explicitly discuss the limitations of gist extraction and evaluates its significant impact on downstream performance through both quantitative ablation studies and qualitative case analyses. Furthermore, to address your concerns more deeply, we have added a comparative experiment on Gist granularity.
> - Discussion of Limitations: We explicitly discussed the limitations of relying on prompted LLMs for gist extraction in Appendix A.7. The original text states: "This process may occasionally produce summaries that are imprecise or fail to capture subtle but crucial nuances... potentially propagating errors". We also proposed future improvements in the paper, specifically developing more robust distillation models.
> - Impact Evaluation on Downstream Performance: We conducted detailed quantitative evaluations of the Gist module in both the main text and appendix:
>   - Ablation Study (Table 4 & Section 5.2.3): On the MuSiQue dataset, removing the Gist module resulted in a decrease in accuracy and a significant increase in retrieval overhead.
>   - Detailed Comparison (Appendix A.2 & Table 5): On the 2Wiki dataset, incorporating the Gist module compared to the "Without Gist" setting not only improved EM by 0.9% (from 62.53% to 63.46%) but also reduced retrieval latency by 13.6%.
>   - Qualitative Analysis (Appendix A.5): We demonstrated how Gist corrects "semantic deviation" through a specific case (Chris Evans' filmography), preventing the retriever from being misled by superficially similar but semantically irrelevant information (e.g., "drumming newcomers"). (Please refer to our response to Reviewer 6Kcf for the detailed case analysis).
>
> W2: We thank you for pointing out the issues regarding evaluation scope and baseline comparisons. We first clarify the experimental setting for LightRAG and then demonstrate EchoRAG's superiority in broader scenarios by introducing a new benchmark, GraphBench, which includes various complex tasks.
>
> Regarding LightRAG's performance in Table 2, we must clarify that to ensure a fair and standardized comparison with existing SOTA methods, we directly cited the LightRAG experimental data reported in the original HippoRAG2 paper in our main table. This was done to maintain consistency in experimental settings rather than due to biases in our reproduction.
>
> To address your concerns about "knowledge-intensive tasks" and "complex reasoning," we conducted a comprehensive evaluation on GraphBench. This benchmark covers two distinct domains, Novel and Medical, and includes high-difficulty tasks: Fact Retrieval, Complex Reasoning, and Contextual Summarize.
>
> The experimental results (Accuracy) are shown below:
> Table 1: GraphBench Evaluation on Novel Dataset
> | Category | Model | Fact Retrieval | Complex Reasoning | Contextual Summarize |
> | :--- | :--- | :--- | :--- | :--- |
> | **Basic RAG** | RAG (w/o rerank) | 58.76 | 41.35 | 50.08 |
> | | RAG (w rerank) | 60.92 | 42.93 | 51.30 |
> | **Graph RAG** | MS-GraphRAG | 49.29 | 50.93 | 64.40 |
> | | HippoRAG | 52.93 | 38.52 | 48.70 |
> | | HippoRAG2 | 60.14 | 53.38 | 64.10 |
> | | LightRAG | 58.62 | 49.07 | 48.85 |
> | | Fast-GraphRAG | 56.95 | 48.55 | 56.41 |
> | | RAPTOR | 49.25 | 38.59 | 47.10 |
> | | Lazy-GraphRAG | 51.65 | 49.22 | 58.29 |
> | | **EchoRAG** | **61.48** | **55.10** | **66.35** |
>
> Table 2: GraphBench Evaluation on Medical Dataset
> | Category | Model | Fact Retrieval | Complex Reasoning | Contextual Summarize |
> | :--- | :--- | :--- | :--- | :--- |
> | **Basic RAG** | RAG (w/o rerank) | 63.72 | 57.61 | 63.72 |
> | | RAG (w rerank) | 64.73 | 58.64 | 65.75 |
> | **Graph RAG** | MS-GraphRAG | 38.63 | 47.04 | 41.87 |
> | | HippoRAG | 56.14 | 55.87 | 59.86 |
> | | HippoRAG2 | 66.28 | 61.98 | 63.08 |
> | | LightRAG | 63.32 | 61.32 | 63.14 |
> | | Fast-GraphRAG | 60.93 | 61.73 | 67.88 |
> | | RAPTOR | 54.07 | 53.20 | 58.73 |
> | | Lazy-GraphRAG | 60.25 | 47.82 | 57.28 |
> | | **EchoRAG** | **73.95** | **74.43** | **74.77** |
>
> EchoRAG achieved a significant advantage in Medical Complex Reasoning (74.43 vs. HippoRAG2 61.98), an improvement of over 12%. In the Contextual Summarize task, EchoRAG also maintained a lead in both domains (Novel: 66.35, Medical: 74.77). This strongly proves that our method is not only suitable for QA but also effective for complex tasks requiring global semantic understanding. Whether in the narrative-heavy Novel domain or the highly specialized Medical domain, EchoRAG demonstrated consistent superiority, validating the effectiveness of the Cognitive Diffusion Module in constructing a global semantic landscape.We will include these extended experiments in the final version to fully demonstrate EchoRAG's robustness across multi-task scenarios.

---

> ### Author Response · Authors · 2025-12-04
>
> Response to Questions (Q1 & Q2)
>
> Q1: Robustness to Gist Extraction Errors
> This issue ("hallucination" or imprecision) is indeed a common challenge with Large Language Models. Therefore, EchoRAG does not rely solely on the gist or triples to generate the final RAG response. As demonstrated in Appendix A.4, we feed the corresponding original passages (retrieved via the gist-guided graph) into the generator. This "Grounding" mechanism significantly mitigates the impact of potential LLM comprehension bias in the gist extraction phase and greatly enhances the quality and factual accuracy of the final response.
>
> Q2: Scalability of the Cognitive Diffusion Module (CDF)
> Regarding the computational complexity of the Cognitive Diffusion Module (CDF) on larger graphs, our analysis is as follows:
> The core of CDF is an iterative update based on sparse matrix multiplication: $\pi_{t+1}=(1-\gamma)W^{\top}\pi_{t}+\gamma\pi_{0}$. Its computational complexity is $O(k \cdot |E|)$, where $k$ is the number of iterations (experimentally observed to be $k < 20$) and $|E|$ is the number of edges in the graph. Since knowledge graphs are typically highly sparse, this process is purely algebraic and does not require invoking an LLM for graph traversal during inference, making the overhead extremely low and scaling linearly with graph size compared to agent-based graph reasoning methods.
>
> Our experimental data strongly supports this. As shown in Table 6, on complex datasets containing massive entities and relations (e.g., 2Wiki and MuSiQue), EchoRAG's total retrieval time (including diffusion) is significantly lower than LightRAG and HippoRAG2. On MuSiQue, EchoRAG took 1700.58s, far lower than LightRAG's 2886.61s. This proves that CDF maintains high computational efficiency even when processing graphs with tens of thousands of nodes and edges.
>
> Beyond algorithmic efficiency, another major advantage of EchoRAG is the effective compression of graph scale via Semantic Gist. Unlike baseline methods (e.g., HippoRAG2) that tend to retain a large number of redundant entity nodes, EchoRAG focuses on core semantics, constructing a leaner and more efficient graph. We compare the number of nodes between EchoRAG and HippoRAG2 across datasets below:
> | Dataset | HippoRAG2 Nodes | EchoRAG Nodes | Reduction |
> | :--- | :--- | :--- | :--- |
> | **PopQA** | 94,420 | **67,123** | **-28.9%** |
> | **NQ** | 96,537 | **84,430** | **-12.5%** |
> | **2Wiki** | 55,663 | **50,311** | **-9.6%** |
> | **MuSiQue** | 113,297 | **101,273** | **-10.6%** |
> | **HotpotQA** | 104,916 | **91,720** | **-12.6%** |
> The data shows that the number of graph nodes constructed by EchoRAG is consistently 10%~30% less than that of HippoRAG2, further contributing to its scalability.

---

### Note · Program_Chairs · 2026-01-17
**Submission Desk Rejected by Program Chairs**

The following references in this submission do not refer to real documents and/or have major errors in bibliographic information:

 Zhaochen Sun, Zhiwei Wang, Haitian Sun, Hao Sheng, and Yizhou Wang. Think-on-graph: Deep and responsible reasoning of large language model on knowledge graph. In Proceedings of the 29th ACM SIGKDD Conference on Knowledge Discovery and Data Mining, 2023.